# Green Synthesis of Selenium Nanoparticles by Grape Seed Extract Synergized with Ascorbic Acid: Preparation Optimization, Structural Characterization, and Functional Activity

**DOI:** 10.3390/foods14173002

**Published:** 2025-08-27

**Authors:** Hua Cheng, Li Wang, Shuqing Jia, Lu Wang, Shuiyuan Cheng, Yingtang Lu, Linling Li

**Affiliations:** 1School of Modern Industry for Selenium Science and Engineering, Wuhan Polytechnic University, Wuhan 430048, China; 15717341028@163.com (L.W.); sqing0222@163.com (S.J.); m15656753051@163.com (L.W.); yingtlu@whu.edu.cn (Y.L.); 2National R&D Center for Se-Rich Agricultural Products Processing, Wuhan Polytechnic University, Wuhan 430023, China; 12316@whpu.edu.cn

**Keywords:** grape seed extract, selenium nanoparticles, response surface methodology, antibacterial, antioxidant

## Abstract

Selenium nanoparticles (SeNPs) have broad application prospects in food preservation and drug development. In this study, grape seed extract (GSE) was used as a natural reducing agent and stabilizer, combined with ascorbic acid (Vc) for the green synthesis of SeNPs (GSE-SeNPs). The preparation process and structural stability were systematically optimized. Response surface methodology (RSM) was employed to optimize the concentrations of Vc and GSE, reaction time, and reaction temperature, aiming to screen out the optimal synthesis conditions with small particle size, good dispersibility, and the smallest PDI value. The results showed that the average particle size of GSE-SeNPs was 74.86 ± 6.07 nm, the PDI was 0.159 ± 0.028, and the Zeta potential was −30.42 ± 0.54 mV, indicating good stability. Characterization results revealed that GSE participated in the reduction and stabilization of SeNPs through electrostatic interactions and hydrogen bonds, forming spherical nanoparticles with a dense structure and good surface organic coating. In addition, GSE-SeNPs exhibited excellent DPPH free radical scavenging ability and antibacterial activity against *Staphylococcus aureus* in vitro. GSE-SeNPs combine green synthesis, structural stability, and multifunctional biological activities, and have the potential to be used as functional nanomaterials in food preservation and safety fields.

## 1. Introduction

The food processing industry generates substantial amounts of biomass by-products annually. Improper disposal not only wastes resources but also imposes severe environmental burdens. In recent years, biomass resource utilization strategies based on circular economy principles have garnered widespread attention. Notably, converting agricultural by-products into high-value-added biomaterials and functional products has emerged as an effective pathway toward sustainable development [1,2]. As a typical fruit and vegetable processing sector, the wine industry produces grape pomace as its primary by-product, consisting of residual grape seeds, skins, and stems. Literature reports indicate that processing 1 kg of fresh grapes yields approximately 0.2 kg of pomace, with grape seeds accounting for roughly 25% of this mass [3,4].

Grape seed extract (GSE), primarily derived from grape seeds and skins, is enriched in polyphenolic bioactive components such as catechin, epicatechin, proanthocyanidins, and resveratrol [5,6]. Extensive research has validated GSE’s potent antioxidant [7], anti-inflammatory [8], and antibacterial activities [9]. Its mechanism of action—as a hydrogen donor, metal chelator, and singlet oxygen scavenger—confers exceptional free radical scavenging capabilities both in vitro and in vivo [10]. Notably, GSE exhibits antibacterial activity, particularly inhibiting Gram-positive bacteria, thus serving as a potential food preservative [11]. Furthermore, polyphenolic compounds in GSE can interact with SeNPs through non-covalent bonding (e.g., hydrogen bonding, electrostatic interactions), thereby enhancing SeNP stability and biological activity [12]. Owing to its wide availability and cost-effectiveness, GSE represents an ideal polyphenols source [10].

In recent years, the green synthesis of selenium nanoparticles (SeNPs) using plant extracts has garnered significant attention due to its environmental benignity and mild reaction conditions [13]. This technology utilizes natural compounds abundant in plant sources (e.g., polyphenols, flavonoids) containing hydroxyl, carboxyl, amino, or aldehyde functional groups to reduce high-valent Se (e.g., sodium selenite, Na_2_SeO_3_) to zero-valent SeNPs (Se^0^) via reduction reactions or coordination complexation [13]. Concurrently, active components in plant extracts stabilize nanoparticles during synthesis through electrostatic interactions, hydrogen bonding, and hydrophobic effects, inhibiting aggregation and sedimentation, while enhancing physicochemical stability and biological functionality [14,15]. Among these approaches, polyphenol-mediated SeNP synthesis—integrating antioxidant and food preservation properties—has demonstrated promising application in natural food additives and intelligent packaging materials [16,17].

This study employs ethanol extraction combined with AB-8 macroporous resin purification to prepare high-purity GSE, which serves as both a green reducing agent and stabilizer. Ascorbic acid (Vc) is synergistically used to reduce Na_2_SeO_3_ for synthesizing GSE-modified selenium nanoparticles (GSE-SeNPs). Under fixed Na_2_SeO_3_ concentration, the effects of Vc concentration, GSE concentration, reaction time, and temperature on SeNP formation are systematically investigated. Dynamic Light Scattering (DLS) and UV–Vis absorption spectroscopy are used to analyze nucleation and growth behaviors, optimizing synthesis parameters. Further characterization of GSE-SeNPs—including particle size distribution, morphological features, surface structure, crystalline properties, and stability mechanisms—is conducted using scanning electron microscopy (SEM), transmission electron microscopy (TEM), energy-dispersive spectroscopy (EDS), Fourier transform infrared spectroscopy (FT-IR), X-ray photoelectron spectroscopy (XPS), X-ray diffraction (XRD), thermogravimetric analysis (TGA), and centrifugal stability analysis. The findings aim to provide theoretical and technical support for constructing GSE-based green functional nanomaterials.

## 2. Materials and Methods

### 2.1. Materials and Reagents

Grape seeds were purchased from Anguo Yaoyuan Trading Co., Ltd. (Baoding, China). Vc, Na_2_SeO_3_, HNO_3_, HCl, absolute ethanol, and petroleum ether were obtained from Sinopharm Chemical Reagent Co., Ltd. (Shanghai, China). Se powder (Se^0^, purity ≥ 99.5%, CAS No. 7782-49-2) was purchased from Shanghai Maclin Biochemical Technology Co., Ltd. (Shanghai, China; No. S817648). Methanol and potassium persulfate were supplied by Tianjin Damao Chemical Reagent Co., Ltd. (Tianjin, China); α-naphthol was purchased from Shanghai Macklin Biochemical Technology Co., Ltd. (Shanghai, China). Macroporous adsorption resin AB-8 was acquired from PANalytical B.V. (Almelo, The Netherlands), and 2,2-diphenyl-1-picrylhydrazyl (DPPH), 2,2′-azino-bis (3-ethylbenzothiazoline-6-sulfonic acid) diammonium salt (ABTS), and ferrous sulfate heptahydrate (FeSO_4_·7H_2_O) were purchased from Sigma-Aldrich (St. Louis, Missouri, USA). Salicylic acid and 30% hydrogen peroxide solution were obtained from Apel Chemical Reagent Co., Ltd. (Shanghai, China).

The *Escherichia coli* strain used in the bacteriostatic experiment is ATCC8739, and the *Staphylococcus aureus* strain is ATCC43300, both of which are preserved by the Microbiology Laboratory of the National Selenium Center, Wuhan Polytechnic University.

### 2.2. Preparation of GSE

Selected grape seeds were freeze-dried, ground using a ball mill, and sieved through a 40-mesh sieve. An appropriate amount of grape seed powder was weighed into a brown glass bottle, mixed with an equal volume of petroleum ether (1:1, *w*/*v*), and oscillated in a shaker (25 °C, 200 rpm) for 36–48 h to remove lipid-soluble impurities. After extraction, the mixture was filtered via Buchner funnel, and the resulting filter cake was dried at 40 °C, ground, and resieved through a 40-mesh sieve for subsequent use.

Defatted grape seed powder was mixed with 60% ethanol solution at a solid-to-liquid ratio of 1:15 (g/mL) and subjected to ultrasonic extraction in an ultrasonic cleaner (45 °C, 200 W, 45 kHz) for 70 min. The extract was centrifuged at 6000 rpm for 20 min at 4 °C, and the supernatant was collected as crude grape seed polyphenol extract, which was refrigerated at 4 °C for storage.

The crude extract was dissolved in ultrapure water and purified using AB-8 macroporous adsorption resin. After thorough pretreatment with ethanol and water, the AB-8 resin was packed into a column using the wet method and equilibrated with ultrapure water to eliminate air bubbles. Samples were loaded at a flow rate of 3 mL/min, followed by standing for a certain period to ensure sufficient resin adsorption. Subsequently, the column was eluted with 5–6 bed volumes of ultrapure water until the Molish reaction of the eluate was negative, indicating the removal of polysaccharide impurities. The target components were then eluted with 80% ethanol solution at a flow rate of 1 mL/min, and the eluate was collected. All eluates were combined, evaporated to concentrate, and freeze-dried to obtain high-purity GSE powder. The final product was sealed, protected from light, and stored at −20 °C for subsequent experiments.

### 2.3. Green Synthesis and Optimization of Preparation Parameters for GSE-SeNPs

In this study, a green reduction method was employed to synthesize grape seed polyphenol-loaded GSE-SeNPs, using GSE and Vc as synergistic reducing agents to reduce Na_2_SeO_3_ to zero-valent Se (Se^0^). The reaction system was treated in a magnetically stirred water bath, and the product was recovered via low-temperature centrifugation, followed by lyophilization to obtain stable red powder. The particle size, dispersibility, and absorption properties were characterized using Dynamic Light Scattering (DLS) and UV–Vis spectroscopy.

To optimize preparation conditions, single-factor experiments were first conducted under a constant Na_2_SeO_3_ concentration (5 mM) to investigate the effects of Vc concentration (1–6 mM), GSE dosage (250–1500 mg/L), reaction time (0.5–2 h), and reaction temperature (25–65 °C) on GSE-SeNP formation. Subsequently, a Box–Behnken design (BBD) was used to construct a four-factor, three-level response surface model, with polydispersity index (PDI) as the response value. Design-Expert 13.0.1.0 software was employed for multiple regression and analysis of variance (ANOVA) to evaluate the significance of individual factors and their interactions. Visual optimization was performed using response surface and contour plots. The optimal parameters for synthesizing GSE-SeNPs were finally screened, providing theoretical support for structural control and application research. The experimental design is shown in Appendix A.

### 2.4. Characterization of GSE-SeNPs

#### 2.4.1. Morphological and Structural Observation

##### TEM

A suspension of GSE-SeNPs was sonicated for 5 min to homogenize. A 5 μL aliquot was dropped onto a copper grid support film and allowed to settle for 8 min. Excess liquid was blotted with filter paper, and the grid was dried overnight in a desiccator. TEM observation was performed to obtain particle morphology and size information.

##### SEM

An appropriate amount of GSE-SeNPs lyophilized powder was fixed on a sample stage with copper conductive adhesive, sputter-coated with gold, and observed via SEM to analyze surface morphology.

#### 2.4.2. Surface Element Composition and Functional Group Analysis

##### SEM-EDX

Based on the SEM sample preparation protocol, energy-dispersive X-ray spectroscopy (EDX) was used to analyze the surface elemental composition of GSE-SeNPs. Combined with SEM images, joint characterization of distribution and structure was performed.

##### FT-IR

Approximately 3 mg of lyophilized GSE-SeNPs sample was mixed with 100 mg of dried anhydrous KBr, thoroughly ground in an agate mortar (particle size ~2 μm), and pressed into pellets (pressure 20 MPa, pressing time 0.5 min). FT-IR analysis was conducted to investigate surface functional groups and chemical structural characteristics.

##### XPS Analysis

Lyophilized GSE-SeNP powder was subjected to XPS measurement. The analysis chamber vacuum was maintained at 5 × 10^−10^ Torr, using a monochromatic Al Kα X-ray source (energy 1486.6 eV, power 5 mA × 15 kV, spot size 700 × 300 μm). Survey scans were performed with a pass energy of 160 eV, and narrow scans with 40 eV, each scanned once. All spectra were energy-calibrated using the C1s peak at 284.6 eV.

##### XRD Analysis

Lyophilized GSE-SeNPs samples were packed into sample cells and flattened. XRD was used for crystal structure analysis, with operating voltage set to 40 kV, current 40 mA, and a continuous step-scanning mode over a 2θ range of 5–90° at a scan rate of 10°/min.

### 2.5. Stability Testing of GSE-SeNPs

#### 2.5.1. Thermal Stability

The thermal stability of GSE-SeNPs was evaluated using a thermogravimetric analyzer (TGA 4000, PerkinElmer, Waltham, MA, USA). Approximately 2 mg of lyophilized sample was placed in an alumina crucible and heated from 30 °C to 600 °C at a rate of 10 °C/min under a continuous flow of high-purity nitrogen (10 mL/min) as the protective gas. Mass loss was recorded to analyze thermal decomposition characteristics.

#### 2.5.2. Centrifugal Stability

The centrifugal stability of GSE-SeNP colloids was assessed using a LUMiFuge analytical centrifuge (LUM GmbH, Berlin, Germany). GSE-SeNP samples in hot solution were injected into specialized tubes and centrifuged at 3000 rpm for 3600 s at 25 °C, with scanning intervals of 10 s. Sedimentation behavior and stability were evaluated by analyzing laser transmittance profiles across height and time dimensions.

### 2.6. In Vitro Functional Evaluation of GSE-SeNPs

To assess the biological activity of the prepared GSE-SeNPs, samples synthesized under single-factor experimental conditions were selected for in vitro antioxidant and antibacterial assays. Antioxidant performance was evaluated via DPPH, ABTS, and hydroxyl radical scavenging capacities [18], while antibacterial activity was determined using the disk diffusion method against *S. aureus* and *E. coli*.

#### 2.6.1. DPPH Radical Scavenging Capacity Assay

A DPPH working solution (0.1 mmol/L in ethanol) was prepared. 100 μL of DPPH solution was added to a microplate well, followed by 20 μL of GSE-SeNP sample solutions at various concentrations. The plate was sealed with aluminum foil to protect from light and incubated at room temperature for 30 min. Absorbance was measured at 517 nm using a microplate reader. Each experiment was performed in triplicate, and the mean value was used for calculations. The DPPH scavenging rate was calculated using the formula:Scavenging rate (%) = (1 − (Ai − Aj)/Ac) × 100%(1)
where Ai is the absorbance of the mixture of sample solution and DPPH, Aj is the absorbance of the sample alone, and Ac is the absorbance of the DPPH solution.

#### 2.6.2. ABTS Radical Scavenging Capacity Assay

ABTS radical solution was generated by reacting ABTS with potassium persulfate and diluted with ethanol to an absorbance of 0.70 ± 0.02 at 734 nm. 200 μL of ABTS working solution and 20 μL of sample solutions at different concentrations were added to each well. Following 6 min of oscillation in the dark, the absorbance of the product was measured at 734 nm. All samples were assayed in triplicate, and the mean value was calculated. The ABTS scavenging rate was calculated as:Scavenging rate (%) = (1 − (Ai − Aj)/Ac) × 100%(2)
where Ai is the absorbance of the mixture of sample solution and ABTS, Aj is the absorbance of the sample alone, and Ac is the absorbance of the ABTS solution.

#### 2.6.3. Hydroxyl Radical Scavenging Capacity Assay

Solutions of FeSO_4_ (9 mmol/L), H_2_O_2_ (8.8 mmol/L), and salicylic acid (9 mmol/L) were prepared separately. In a microcentrifuge tube, 50 μL of sample solution, 50 μL of FeSO_4_ solution, and 100 μL of H_2_O_2_ solution were sequentially added, mixed, and incubated for 10 min. 50 μL of salicylic acid solution was then added, and the mixture was vortexed again and incubated at room temperature for 30 min. Absorbance was measured at 510 nm, with triplicate determinations for each sample. The scavenging rate was calculated using the formula:Scavenging rate (%) = (1 − (Ai − Aj)/Ac) × 100%(3)
where Ai is the absorbance of the mixture of sample solution and •OH reaction system, Aj is the absorbance of the sample solution alone, and Ac is the absorbance of the •OH reaction system.

#### 2.6.4. Antibacterial Activity Assay

The disk diffusion method was used to evaluate the in vitro antibacterial activity of GSE-SeNPs against pathogenic *S. aureus* and *E. coli*. Activated bacterial strains were inoculated into LB liquid medium and cultured overnight at 37 °C, then diluted to approximately 10^6^ CFU/mL. 20 μL of bacterial suspension was uniformly spread onto pre-prepared LB agar plates to form a bacterial lawn. Sterile filter paper disks (6 mm diameter) were impregnated with GSE-SeNPs, Na_2_SeO_3_, Vc, and GSE solutions, air-dried, and placed in separate regions of the inoculated plates. The treated plates were incubated upside down at 37 °C for 12 h. The diameter of inhibition zones around the disks was measured using vernier calipers to assess antibacterial efficacy. All experiments were performed in triplicate.

### 2.7. Data Processing and Statistical Analysis

All measurements were independently repeated at least three times under identical conditions. Data were subjected to analysis of variance (ANOVA) for significance testing, and differences were compared using Duncan’s multiple range test, performed using SPSS 26.0 software (IBM Corp., Chicago, IL, USA). A significance level of *p* < 0.05 was considered statistically significant. Graphs and data visualization were generated using Origin 2021 (OriginLab Corp., Northampton, MA, USA) and GraphPad Prism 8 (GraphPad Software, San Diego, CA, USA). All results are expressed as Mean ± Standard Error (Mean ± SE).

## 3. Results and Discussion

### 3.1. Optimization of GSE-SeNP Synthesis Process

Single-factor optimization results showed that under conditions of 5 mM Vc, 500 mg/L GSE, 90 min reaction time, and 45 °C temperature, GSE and Na_2_SeO_3_ synergistically formed GSE-SeNPs with small particle size, low PDI, and good dispersibility, exhibiting excellent structural uniformity and colloidal stability [19] (Appendix A).

Based on these findings, a BBD combined with Design-Expert 13.0.1.0 software was further employed to establish a multivariate quadratic regression model for four key variables: Vc concentration (X_1_), GSE dosage (X_2_), reaction time (X_3_), and reaction temperature (X_4_), with PDI as the response value (Y). Experimental data are provided in Appendix A Appendix A.

Analysis of variance (ANOVA) results (Appendix A) indicated high overall model fitness (*p* < 0.001) with no significant lack of fit (*p* = 0.1603), verifying the reliability and predictive ability of the regression equation. Further analysis revealed that all four variables and selected interaction terms exerted statistically significant effects on PDI values, preliminarily establishing the role of each factor in regulating nanostructural stability. These findings provide theoretical support and parameter basis for the precise synthesis of highly dispersible and uniform GSE-SeNPs.

Based on the three-dimensional response surface and contour plots derived from the quadratic polynomial regression model, the interactions between X_1_ × X_4_ (Vc concentration and reaction temperature, Figure 1a) and X_3_ × X_4_ significantly affected PDI (reaction time and reaction temperature, Figure 1b). PDI initially decreased with increasing temperature, followed by an increase, presumably related to Vc’s thermal instability and degradation. Similarly, the synergistic increase in reaction time and temperature showed a “first decrease then increase” trend, indicating that time and temperature must be co-optimized to enhance colloidal stability. Other interactions (X_1_ × X_3_ (Figure 1c), X_2_ × X_4_ (Figure 1d) had weaker effects, while X_1_ × X_2_ (Figure 1e) and X_2_ × X_3_ (Figure 1e) were essentially non-significant.

The optimal synthesis conditions for GSE-SeNPs were ultimately determined as: Vc concentration 5 mM, GSE dosage 500 mg/L, reaction time 1.5 h, and reaction temperature 45 °C. Baluken et al. optimized green coffee bean extract and green-synthesized selenium nanoparticles using the Box–Behnken experimental design response surface methodology. This method produced spherical nanoparticles of approximately 100 nm, which have an amorphous (non-uniform) structure and can maintain stability at high temperatures [19,20]. Additionally, in the study of blueberry pomace polyphenols–SeNPs, the results of infrared spectroscopy combined with ultraviolet experiments indicated that there are hydrogen bond-like interactions between the O-H groups of polyphenols and flavonoids and the Se atoms of SeNPs, which helps to improve the stability and dispersibility of TP-SeNPs [21]. These findings provide a theoretical basis for constructing stable and efficient GSE-SeNP systems. Subsequent characterization and functional experiments were conducted under these optimal conditions.

### 3.2. Particle Size Distribution and Morphological Analysis

As shown in Figure 2, GSE-SeNPs appeared as an orange-red transparent suspension in aqueous solution without precipitation or stratification, indicating good dispersibility and stability (Figure 2a). DLS results revealed an average particle size of 74.86 ± 6.07 nm, PDI of 0.159 ± 0.028, and Zeta potential of −30.42 ± 0.54 mV (Table 1). A smaller particle size helps improve the bioavailability and stability of nanoparticles [22], while a lower PDI value indicates a more concentrated particle size distribution [23]. A high negative Zeta potential on the surface of nanoparticles can increase electrostatic repulsion between them, preventing aggregation and thereby enhancing their dispersibility and stability in solution [24]. These results are crucial for the accessibility, stability, and application potential of GSE-SeNPs.

SEM was used to examine the morphology of GSE-SeNPs. At high magnification, the size of the nanoparticles was more clearly visible, appearing approximately spherical with smooth surfaces (Figure 2b). At low magnification, the overall distribution of the nanoparticles could be observed, showing uniform particle size with no obvious aggregation. The analysis results of the SEM images were consistent with the particle size distribution curve. TEM observations revealed that GSE-SeNPs were spherical in shape, uniformly dispersed in solution with no obvious aggregation. These microstructural observations are consistent with the particle size distribution and PDI values in the DLS results, further confirming the uniformity and stability of the nanoparticles. Similar observations were reported for Usnea longissima polysaccharide-modified SeNPs, where structural stability was attributed to non-covalent interactions (CO⋯Se or OH⋯Se) [25]. Collectively, GSE-SeNPs exhibit controllable particle size and consistent morphology, making them suitable for constructing functional food nanocarriers.

### 3.3. Surface Element Composition and Structural Characteristic Analysis

As shown in Figure 3, combined EDS and FT-IR characterization verified the elemental composition and functional group changes of GSE-SeNPs. EDS analysis can accurately detect the elemental composition of GSE-SeNPs and confirm the existence form and content of Se in GSE-SeNPs. The EDS analysis results revealed that carbon (C), oxygen (O), and Se elements were detected at both points (Table 2). The Se contents at the two points were 78.65% and 77.34%, respectively, which were significantly higher than other elements (Table 2). This indicates that Se is the main component of GSE-SeNPs. The presence of C and O suggests that GSE was successfully bound to the surface of SeNPs [26,27].

The FT-IR spectra showed the characteristic bands of GSE, Vc, Na_2_SeO_3_, and GSE-SeNPs. The characteristic bands of GSE were observed at 3380.02 cm^−1^ (O-H stretching vibration), 1612.5 cm^−1^ (C=C stretching vibration), 1512.22 cm^−1^ (C-O stretching vibration), 1445.23 cm^−1^ (C-H bending vibration), and 1285.92 cm^−1^ (C-O stretching vibration) [28]. In GSE-SeNPs, some bands exhibited redshift: 3380.02 cm^−1^ shifted to 3371.9 cm^−1^, 1612.5 cm^−1^ shifted to 1602.55 cm^−1^, 1445.23 cm^−1^ shifted to 1436.27 cm^−1^, and 1285.92 cm^−1^ shifted to 1280.29 cm^−1^ (Figure 3). These band changes indicate chemical interactions (such as the formation of hydrogen bonds or other chemical bonds) between the phenolic hydroxyl groups in GSE extract and Se [29]. This aligns with mechanisms of Se⋯H bonding and functional group displacement reported in relevant literature [30,31]. Furthermore, characteristic bands of Vc and Na_2_SeO_3_ were significantly attenuated in GSE-SeNPs, confirming their reduction to Se^0^ during synthesis. GSE effectively coats and stabilizes SeNPs, endowing them with excellent structural integrity and colloidal stability, supporting their application in functional materials.

### 3.4. Particle Charge Characteristic Analysis

XPS analysis further revealed the elemental valence states and chemical environment of GSE-SeNPs (Figure 4). The full-scan spectrum displayed distinct characteristic peaks for Se, C, and O, confirming their composition as GSE-coated SeNPs, which was highly consistent with EDS results (Figure 4a). The high-resolution Se 3d spectrum showed two peaks at 54.5 eV and 53.8 eV corresponding to Se 3d_3_/_2_ and Se 3d_5_/_2_, respectively (Figure 4b), confirming the presence of Se in the zero-valent state (Se^0^) [32]. No signals for Se^4+^ or Se^6+^ were detected, indicating high selectivity and purity of the reduction process. These results are consistent with studies by Tendenedzai et al. [33] and other plant bioactive-mediated SeNP syntheses [13], demonstrating that GSE serves dual roles as an efficient reducing agent and stabilizer during synthesis. Se in GSE-SeNPs exists stably in a single Se^0^ state, reflecting excellent chemical purity and structural stability, which lays a foundation for their application in food and bioactive materials.

### 3.5. Crystalline Characteristic Analysis

XRD analysis was conducted to reveal the crystal structure characteristics of GSE-SeNPs and their interaction with GSE (Figure 5). Elemental Se exhibits characteristic diffraction peaks at 23.54° and 29.84° (Figure 5a), indicating the presence of crystalline Se [34]. However, the diffraction peak at 29.84° disappears for GSE-SeNPs, and the peak at 23.54° weakens and broadens (Figure 5b), suggesting that GSE binds with SeNPs to form an amorphous crystal. This may be because polyphenol molecules act as a framework, decorating the surface of SeNPs and disrupting the crystal structure, leading to the formation of an amorphous solid. The same conclusion was found in a previous study where blueberry pomace polyphenols were used as a soft template to prepare nano-selenium [31]. Similar observations of diminished or absent diffraction peaks in SeNPs synthesized using *Carica papaya* extract and *Penicillium chrysogenum* have been attributed to the negative charge of flavonoids and phenolic compounds in the extracts [35,36]. Polar groups in GSE may disrupt the ordered arrangement of Se atoms through hydrogen bonding, electrostatic, or complexation interactions [37]. Combined with the FT-IR peak redshift, C/O element coating in EDS, and single Se^0^ valence state in XPS from this study, these results further confirm that GSE acts as both a reducing agent and stabilizer in SeNP synthesis. GSE-SeNPs primarily exhibit an amorphous structure. The negatively charged extract chelation provides enhanced uniformity, stability, and colloidal dispersibility to the nanoparticles, highlighting their potential applications in functional foods and bioactive materials.

### 3.6. Thermal and Centrifugal Stability Evaluation of GSE-SeNPs

The stability of GSE-SeNPs was evaluated using TGA and the LUMiFuge^®^111 system. TGA results showed negligible mass change for GSE-SeNPs from room temperature to 200 °C, with slow pyrolysis occurring between 200 and 500 °C. Weight loss accelerated beyond 500 °C but remained significantly outperforming pure Se (Figure 6a). In contrast, pure Se underwent decomposition between 420 and 500 °C, indicating poor thermal stability. These findings align with Tao et al.’s report on Moringa polysaccharide-embedded SeNPs, which emphasized the critical role of polysaccharide–SeNP composite structures in thermal protection [32]. Furthermore, Zhou et al. prepared SM-EGCG-SeNPs via synergistic capping with starch microgel and tea polyphenol EGCG [38]. The TGA curve also showed significantly slowed weight loss in the 200–500 °C range (Figure 6b), demonstrating excellent thermal decomposition inhibition compared to uncapped SeNP controls—further validating the effectiveness of polyphenols in constructing thermally stable interfacial structures.

Accelerated sedimentation results revealed an instability index of 0.019, far below the 0.1 threshold (Figure 6c), with slow changes in instability curves (Figure 6d) and uniform light transmittance distribution (Figure 6e). No significant aggregation or sedimentation was observed, indicating good colloidal stability. This performance is primarily attributed to the stable network structure formed via hydrogen bonding and electrostatic interactions between polyphenols in GSE and SeNP surfaces, effectively inhibiting particle aggregation [39,40]. Thus, GSE-SeNPs exhibit both excellent colloidal and thermal stability. The polyphenol-constructed organic–inorganic interface not only enhances structural protection but also provides a solid foundation for applications in thermal processing and functional packaging.

### 3.7. Antioxidant and Antibacterial Activity Evaluation

Previous studies have demonstrated that GSE-SeNPs possess good dispersibility and stability. To further verify their functional properties, in vitro antioxidant and antibacterial activities were evaluated. GSE exhibited the strongest scavenging activity against DPPH, ABTS, and •OH, with scavenging rates of 79.2%, 98.5%, and 78.7%, respectively. The scavenging rates of GSE-SeNPs against DPPH, ABTS, and hydroxyl radicals (•OH) were 73.3%, 33.2%, and 51.3%, respectively, which were slightly lower than those of GSE but significantly higher than those of elemental Se (Figure 7). The difference in activity between the nanoparticles and the extract is believed to be due to the consumption of some molecules with antioxidant potential during the synthesis of SeNPs for the reduction reaction of the nanoparticles [26]. This antioxidant effect may be attributed to the abundant phenolic hydroxyl groups in GSE, which act as hydrogen donors to effectively neutralize free radicals [10], while SeNPs enhance synergistic scavenging capacity through electron transfer pathways. Similar results were observed in Ge et al.’s study on *Rosa roxburghii* polyphenol-modified SeNPs, where free radical scavenging rates were also significantly higher than control groups [30].

In the antibacterial experiment, the antibacterial effect of GSE-SeNPs against *S*. *aureus* was significantly better than that against *E. coli*. Both Na_2_SeO_3_ and GSE-SeNPs showed strong inhibitory effects on *S. aureus*, with obvious inhibition zones formed in the experiment, with diameters of 17.43 ± 0.68 mm and 12.62 ± 0.64 mm, respectively. Their antibacterial effects were significantly better than those of Vc or GSE (Figure 7d,e). In contrast, GSE-SeNPs had a stronger inhibitory effect on *E. coli* than Na_2_SeO_3_, with inhibition zones of 11.71 ± 0.58 mm and 5.34 ± 0.23 mm, respectively. No obvious inhibition zones were observed for Vc or GSE against *E. coli*. Salem et al. reported that green SeNPs synthesized using orange peel extract exhibit good antibacterial activity against multidrug-resistant bacteria including *S. aureus*, and the mechanism involves disrupting cell membrane permeability, inducing intracellular ROS accumulation, and interfering with cellular metabolism [41]. Multiple studies have shown that SeNPs exhibit different antibacterial activities against Gram-positive and Gram-negative bacteria. Studies have found that SeNPs have a more significant antibacterial effect on Gram-negative bacteria compared with Gram-positive bacteria [42].

GSE-SeNPs demonstrate excellent functional performance in vitro. As drug delivery carriers, their high surface area and biocompatibility advantages have been reported for targeted delivery of anti-infective and anticancer drugs, helping enhance efficacy while reducing toxic side effects [43,44]. In the functional food field, green-synthesized SeNPs (e.g., from microalgae supplements) have improved animal immunity and growth performance, indicating great potential as nutritional supplements [45]. For food preservation applications, strawberries and peppers coated with SeNPs have been shown to extend shelf life, slow respiration rates, and reduce oxidative damage, supporting their use in intelligent packaging or composite preservation coating [46]. Furthermore, regarding sustained release of SeNPs fertilizer, studies have shown that SeNPs achieve controlled release in soil, promote microbial growth, and improve crop absorption efficiency—an environmentally friendly sustained-release fertilizer strategy [47]. Thus, GSE-SeNPs, with their stable nanostructure and versatile applicability, hold broad prospects for applications in medicine, food, and agriculture.

## 4. Conclusions

This study achieved the green synthesis of GSE-SeNPs using GSE as both reducing agent and stabilizer, synergized with ascorbic acid. Optimized via response surface methodology, the optimal conditions (5 mM Vc, 500 mg/L GSE, 1.5 h, 45 °C) yielded GSE-SeNPs with uniform spherical morphology, average particle size of 74.86 ± 6.07 nm, PDI of 0.159 ± 0.028, and Zeta potential of −30.42 ± 0.54 mV, indicating excellent dispersibility and stability. Structural analysis confirmed GSE interacts with SeNPs via electrostatic forces and hydrogen bonding, forming a protective organic coating that enhances thermal stability (resistant to 200 °C) and anti-aggregation properties. XPS and XRD verified zero-valent Se^0^ in an amorphous structure due to GSE modification. Functional evaluations showed GSE-SeNPs exhibit strong antioxidant activity (73.3% DPPH, 93.1% ABTS, 69.2% hydroxyl radical scavenging) and significant antibacterial activity against *S*. *aureus*. These findings highlight GSE-SeNPs as a promising functional nanomaterial for food preservation, active packaging, and functional food development, supporting sustainable nanotechnology in the food industry.

## Figures and Tables

**Figure 1 foods-14-03002-f001:**
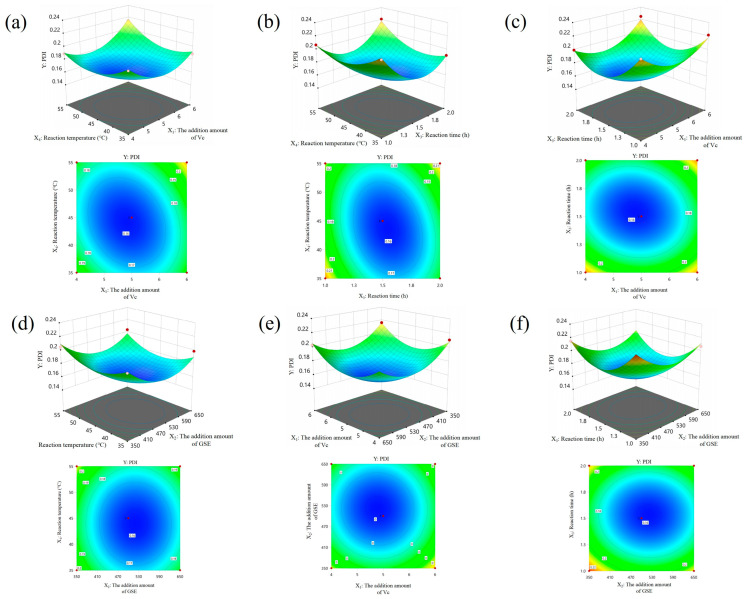
Three-dimensional response surface and corresponding contour plots illustrating the interaction effects of experimental factors on the PDI value (Y), with other variables held at their central levels. (**a**) Interaction between Vc concentration (X_1_) and GSE concentration (X_2_); (**b**) interaction between Vc concentration (X_1_) and reaction time (X_3_); (**c**) interaction between Vc concentration (X_1_) and reaction temperature (X_4_); (**d**) interaction between GSE concentration (X_2_) and reaction time (X_3_); (**e**) interaction between GSE concentration (X_2_) and reaction temperature (X_4_); (**f**) interaction between reaction time (X_3_) and reaction temperature (X_4_).

**Figure 2 foods-14-03002-f002:**
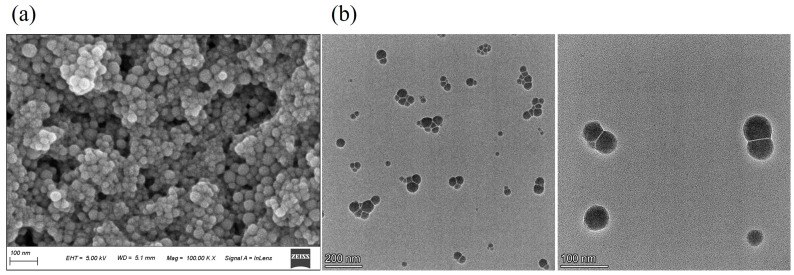
Characterization of particle size and morphology of GSE-SeNPs. (**a**) SEM image at 100 k magnification; (**b**) TEM images at different magnifications.

**Figure 3 foods-14-03002-f003:**
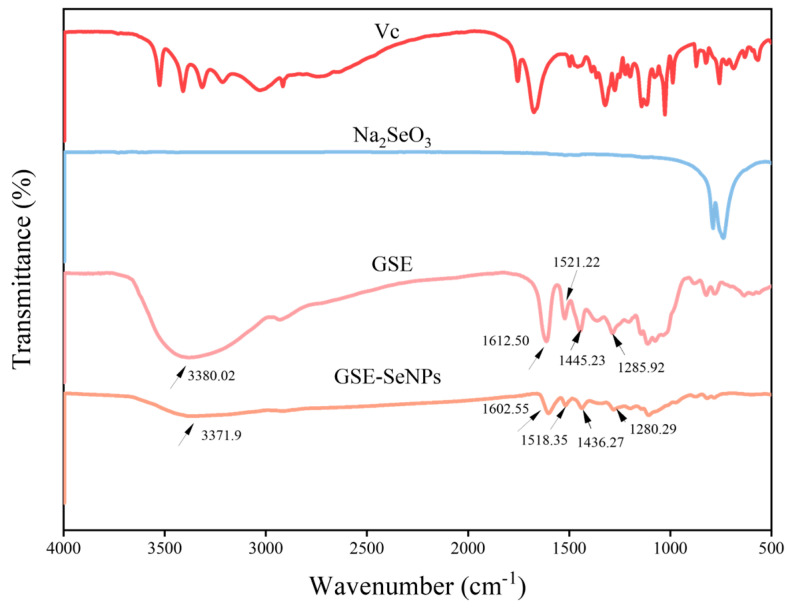
FT-IR spectra of GSE, Vc, Na_2_SeO_3_, and GSE-SeNPs.

**Figure 4 foods-14-03002-f004:**
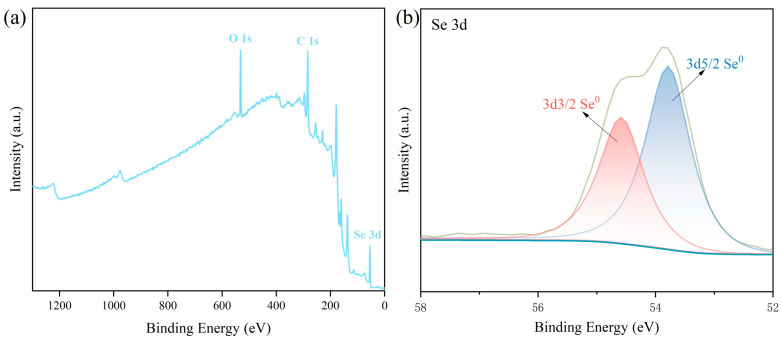
XPS Spectra of GSE-SeNPs. (**a**) Survey spectrum showing the presence of C, O, and Se elements; (**b**) high-resolution spectrum of Se 3d region.

**Figure 5 foods-14-03002-f005:**
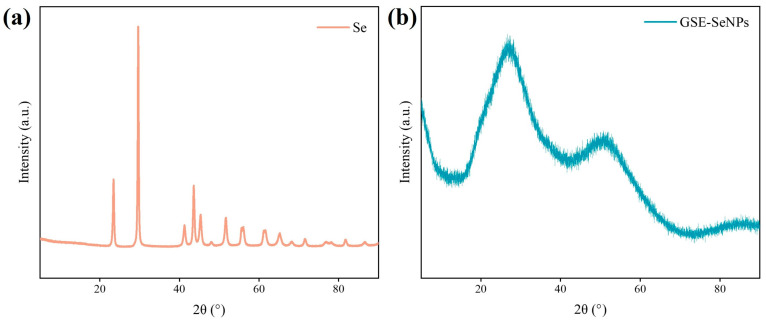
XRD spectra of GSE-SeNPs and Se. (**a**) Elemental Se powder (Se^0^, ≥99.5%, CAS No. 7782-49-2) purchased from Macklin Biochemical Co., Ltd. (Shanghai, China); (**b**) GSE-synthesized nano-Se.

**Figure 6 foods-14-03002-f006:**
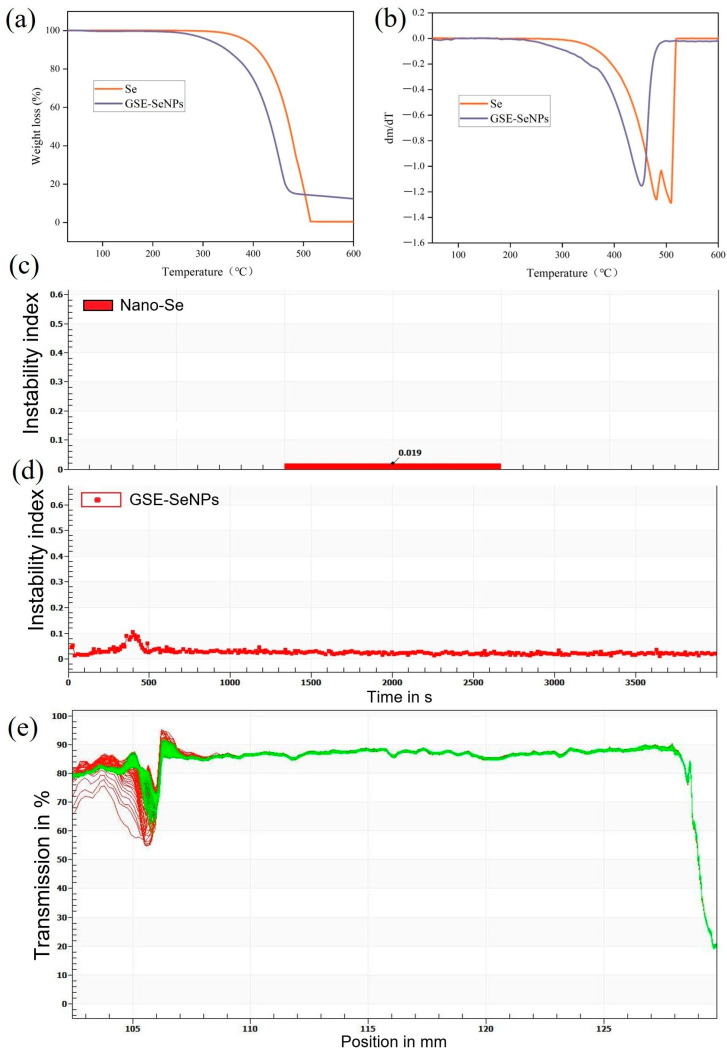
Thermal and centrifugal stability analysis of GSE-SeNPs. (**a**) TGA curves of GSE-SeNPs and Se; (**b**) corresponding DTG (first derivative) curves; (**c**) instability index value of GSE–SeNPs obtained from centrifugal analysis; (**d**) instability index curve showing temporal changes during centrifugation; (**e**) light transmission curve during centrifugation.

**Figure 7 foods-14-03002-f007:**
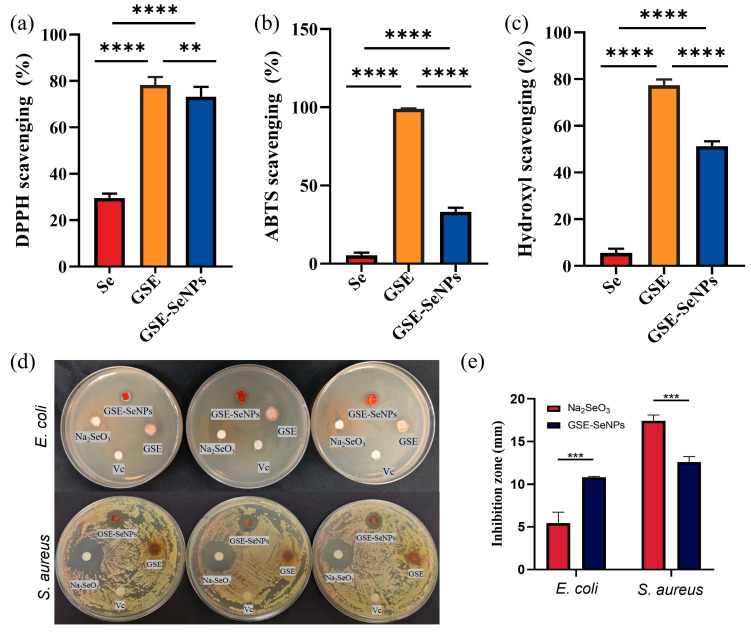
Antioxidant and antibacterial evaluation of GSE-SeNPs. (**a**) DPPH scavenging activity; (**b**) ABTS scavenging activity; (**c**) •OH radical scavenging activity; (**d**) inhibition zones against *E. coli* and *S. aureus* for different treatments; (**e**) inhibition zones of *E. coli* and *S. aureus* treated with Na_2_SeO_3_, GSE–SeNPs, GSE, and Vc. The asterisks indicate statistical significance of differences, with **, ***, and **** representing *p* < 0.05, *p* < 0.01, and *p* < 0.001, respectively.

**Table 1 foods-14-03002-t001:** DLS results of GSE-SeNPs.

Sample Name	Average Particle Size (nm)	PDI	Zeta Potential (mV)
GSE-SeNPs	74.860 ± 6.071	0.159 ± 0.028	−30.417 ± 0.538

**Table 2 foods-14-03002-t002:** Elemental composition of GSE-SeNPs.

Element	Spectrum 23			Spectrum 24		
	wt.%	wt.% Sigma	At%	wt.%	wt.% Sigma	At%
C	17.06	0.47	52.91	18.08	0.46	54.34
O	4.29	0.15	9.99	4.57	0.15	10.31
Se	78.65	0.46	37.10	77.34	0.45	35.35
Total	100			100		

## Data Availability

The original contributions presented in this study are included in the article/Appendix A. Further inquiries can be directed to the corresponding authors.

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
