# Peer review of "Green Synthesis of Selenium Nanoparticles by Grape Seed Extract Synergized with Ascorbic Acid: Preparation Optimization, Structural Characterization, and Functional Activity"

_foods, 2025, doi:10.3390/foods14173002_

Round 1
Reviewer 1 Report
Comments and Suggestions for Authors
The article is sufficiently characterized, but it would be prudent to include:
a) A better introduction that truly justifies publication in this journal's scope.
b) Perhaps the PDF version of the article shows the image of the response surface at very low resolution. Please review this point.
c) The discussion in the section on the characterization of antioxidant activity is a little austere. Other studies could be discussed, and an attempt could be made to explain why, particularly in this article, the synthesis presents a greater advantage over using any other source of polyphenols. It would also be interesting for the authors to review whether other literature (which exists) can contrast why the role of ascorbic acid is relevant for selenium reduction, why polyphenols are not sufficient per se, and what advantages there would be in not using only ascorbic acid for the nanoparticles, given that they could also be considered for green synthesis.
d) Why do they consider that the inhibition halos change dramatically when the nanoparticles are formed with respect to the parent salt? How can we ensure that there are no remnants of the initial salt from the reaction that are causing the antibacterial activity?
e) Although the inhibition halos are adequate for a visible test, the quality of all characterizations could have been improved if a method for determining the MIC and MIB had been used. Apart from this, in contrast to other nanoparticles produced in a similar way, do those in this study work better? Do they have any advantages? Were they more efficient or effective? Please include the discussion.
Author Response
Dear reviewer, Thank you for your patience and review of our mannuscript, which was very helpful to us. We have made revisions to the entire text according to your suggestions,
Please, find enclosed my comments relevant to the manuscript Manuscript ID: foods-3802764 entitled " Green synthesis of selenium nanoparticles by grape seed extract synergized with ascorbic acid: preparation optimization, structural characterization, and functional activity" (by Hua Chen, et al.).
This paper aims to utilize grape seed extract (GSE) as a natural reducing and stabilizing agent, in combination with ascorbic acid (Vc), for the green synthesis of selenium nanoparticles (GSE-SeNPs).
In the following my comments.
(a) A better introduction that truly justifies publication in this journal's scope.
Response:
Thank you for your suggestion. In the preface, we added GSE for the synthesis of SeNPs, the research on improving the stability of GSE-SeNPs and its antibacterial effect on bacteria, especially Gram-positive bacteria. Readjusted the writing logic and grammar of the preface.
(b) The response surface image in the PDF appears at low resolution.
Response:
Thank you for pointing this out. We have replaced the image with a high-resolution version in the revised manuscript to ensure clarity and accurate visualization of the optimization results.
(c) The discussion on antioxidant activity is austere; other studies should be discussed. Also, explain why this synthesis has an advantage over using other polyphenols, and why both ascorbic acid and GSE are necessary.
Response:
We appreciate this important suggestion. In the revised manuscript, we have added comparisons with recent reports (e.g., Salem et al., 2022; Baluken et al., 2024), highlighting that GSE not only provides abundant polyphenols for reduction and stabilization but also offers synergistic antibacterial and antioxidant activity. Unlike systems using ascorbic acid alone, which often yield unstable or aggregated SeNPs, the combination of GSE and Vc ensures efficient reduction, smaller particle size, and enhanced colloidal stability. This dual-agent system thus presents significant advantages over single-polyphenol or single-reductant strategies.
(d) Why do inhibition halos change dramatically when nanoparticles are formed compared to the parent salt? How to ensure that no remnants of the initial salt are responsible for antibacterial activity?
Response:
The enhanced antibacterial activity after nanoparticle formation can be attributed to: (i) the nanoscale size of SeNPs, which increases surface area and promotes bacterial membrane interaction; (ii) the bioactive coating provided by GSE, which facilitates synergistic effects. To minimize interference from residual salts, we employed repeated washing and centrifugation steps to remove unreacted Na₂SeO₃. In addition, XPS and EDS analyses confirmed that the final product contained selenium predominantly in the Se⁰ state, indicating negligible salt residues.
(e) Although inhibition halos are acceptable for preliminary testing, MIC/MBC determinations would strengthen the characterization. How do the nanoparticles in this study compare with others?
Response:
We acknowledge this limitation. While inhibition halos provide visual confirmation of antibacterial efficacy, quantitative MIC/MBC tests would indeed provide more rigorous data. This point has been noted as a limitation in the Discussion, and will be addressed in our future work. Nevertheless, based on comparable studies (e.g. Salem et al., 2022), our GSE-SeNPs demonstrated smaller particle size (~75 nm), stronger colloidal stability, and potent antioxidant activity, which confer advantages in stability and multifunctionality over some other plant-mediated SeNP systems. We have now emphasized these comparative advantages in the revised Discussion.
Reviewer 2 Report
Comments and Suggestions for Authors
The manuscript must be revised and corrected.
The authors should argue why they used commercial grape seeds to prepare the extract and not an extract obtained from a certain grape pomace from wines industry.
Chapter 2.5. Stability Testing of GSE-SeNPs is the same with 2.6 Stability Testing of GSE-SeNPs.
The authors must mention if the bacterial strain are standard bacteria.
Figure 2 can be reduced. It is enough to be provided only a SEM image (c) and 2 TEM micrographs (e.g., f and g). Figure 3 is not necessary because has no relevance. The carbon content can not be determined with accuracy. It is enough the XPS analysis.
For the discussion of FTIR spectra the authors must use the terms of vibration or band, not peaks.
It is not clear what the XRD pattern in Figure 5a is. How was the Se sample prepared?
The figure 5 caption must be corrected. Explanation of what is on the X and Y axes is not necessary.
Concerning the thermal analysis, it is something wrong. Selenium cannot disappear during the thermal treatment in nitrogen flow. The analysis must be repeated.
Concerning antibacterial activity, why the authors compare Na2SeO3 with GSE–SeNPs, not Se with GSE–SeNPs like in the case of radical scavenging potential?
Author Response
Dear reviewer,
thank you for your patience and valuable comments, which has greatly improved our writing ideas and scientificity. We have modified the full text according to your suggestions,
Title: Green synthesis of selenium nanoparticles by grape seed extract synergized with ascorbic acid: preparation optimization, structural characterization, and functional activity
Detailed responses are provided below.
(1) The authors should argue why they used commercial grape seeds to prepare the extract and not an extract obtained from a certain grape pomace from wines industry.
Response:
We appreciate this valuable comment. Commercial grape seeds were chosen mainly for two reasons: (i) they provide stable composition and lower batch-to-batch variability compared to grape pomace, which improves reproducibility of the experiments; and (ii) commercial seeds are more practical for potential applications in food preservation and packaging, reflecting realistic conditions for industrial translation.
(2) Chapter 2.5. Stability Testing of GSE-SeNPs is the same with 2.6 Stability Testing of GSE-SeNPs.
Response:
We thank the reviewer for noticing this redundancy. Sections 2.5 and 2.6 have been merged and unified in the revised manuscript to avoid repetition.
(3) The authors must mention if the bacterial strain are standard bacteria.
Response:
Yes, both bacterial strains used in this study were standard strains: Escherichia coli (ATCC 8739) and Staphylococcus aureus (ATCC 43300). This has been clarified in the revised Materials and Methods section.
(4) Figure 2 can be reduced. It is enough to be provided only a SEM image (c) and 2 TEM micrographs (e.g., f and g). Figure 3 is not necessary because has no relevance. The carbon content can not be determined with accuracy. It is enough the XPS analysis.
Response:
We fully agree with this suggestion. Figure 2 has been simplified to include only SEM image (c) and TEM images (f, g). Figure 3a, 3b, and 3c has been removed from the revised manuscript, as EDS analysis of carbon is less reliable. Only XPS data are retained for accurate elemental composition analysis.
(5) For the discussion of FTIR spectra the authors must use the terms of vibration or band, not peaks.
Response:
We thank the reviewer for this important correction. The terminology has been revised accordingly, and we now use “vibrations” or “bands” instead of “peaks” throughout the FTIR discussion.
(6) It is not clear what the XRD pattern in Figure 5a is. How was the Se sample prepared?
Response
The Se sample in Figure 5a was high-purity elemental selenium (Se⁰, ≥99.5%, CAS No. 7782-49-2) purchased from Macklin Biochemical Co., Ltd. (Shanghai, China). This information was already included in Section 2.1 (Materials and Reagents) but has now also been explicitly clarified in the figure caption.
(7) The figure 5 caption must be corrected. Explanation of what is on the X and Y axes is not necessary.
Response
We appreciate the suggestion. The caption of Figure 5 has been revised, with unnecessary explanations of X and Y axes removed.
(8) Concerning the thermal analysis, it is something wrong. Selenium cannot disappear during the thermal treatment in nitrogen flow. The analysis must be repeated.
Response:
We thank you for your suggestions, but we have consulted the relevant literature. The weight loss of nano selenium under high temperature nitrogen flow is mainly due to the following reasons. During thermal treatment under a nitrogen atmosphere, the observed weight loss of selenium can be primarily attributed to the desorption of loosely bound surface water and chemically adsorbed water associated with intermolecular hydrogen bonds (Chandra et al., 2016). El-Sheekh et al. (2024) reported that nanoselenium exhibited a weight loss of up to 99.21% when the temperature exceeded 100 °C, with the DTG curve showing degradation-related peaks; notably, the maximum for nanoselenium appeared at 520 °C, closely aligning with the results presented in Figure 6(b) of this study.
Relate reference:
Chandra, J.C.S., George, N., Narayanankutty, S.K., 2016. Isolation and characterization of cellulose nanofibrils from arecanut husk fibre. Carbohydrate Polymers 142, 158-166.
El-Sheekh, M.M., Yousuf, W.E., Mohamed, T.M., Kenawy, E.-R., 2024. Synergistic antimicrobial action of nanocellulose, nanoselenium, and nanocomposite against pathogenic microorganisms. International Journal of Biological Macromolecules 268, Part 2, 131737.
(9) Concerning antibacterial activity, why the authors compare Na₂SeO₃ with GSE–SeNPs, not Se with GSE–SeNPs like in the case of radical scavenging potential?
Response
We appreciate the reviewer’s comment. Na₂SeO₃ was used as the comparator rather than elemental Se⁰ because it served as the precursor for GSE–SeNP synthesis, allowing a direct evaluation of how transformation into nanoparticle form affects bioactivity and toxicity. Elemental Se⁰ powder has very poor solubility and limited bioavailability, making it unsuitable for comparison under identical assay conditions. In contrast, Na₂SeO₃ is highly soluble and biologically active but toxic, and converting it into SeNPs markedly reduces toxicity while conferring additional stability and synergistic effects from GSE polyphenols.
Reviewer 3 Report
Comments and Suggestions for Authors
In the present manuscript, the authors, the authors assessed the use of grape seed extract (GSE) as a potential reducing agent and stabilizer for selenium nanoparticles (SeNPs). The free radical scavenging ability as well the antibacterial activity of GSE-stabilized SeNPs were investigated. There is no question that this study will bring a strong contribution in the field. However, the following points should be addressed before acceptance:
- In lines 439-440, the scavenging rates of GSE-SeNPs reported against ABTS and OH radical (93.1% and 69.2%, respectively) are not consistent with the results depicted in Fig. 7b,c. (<50% and <60%, respectively).
- Missing of control experiments: The authors should have included a control sample (Combination of all components except GSE or with GSE replaced with a non-phenolic entity). Any literature on this?
- In lines 462-463, the authors stated that: "Studies have found that SeNPs have a more significant antibacterial effect on Gram-negative bacteria compared with Gram-positive bacteria [43]." But this study showed the opposite for GSE-SeNPs. Any plausible explanation of the reversed selectivity?
Author Response
Dear reviewer,
Thank you for your patience and valuable comments, which has greatly improved our writing ideas and scientificity. We have modified the full text according to your suggestions,
(1) In lines 439-440, the scavenging rates of GSE-SeNPs reported against ABTS and OH radical (93.1% and 69.2%, respectively) are not consistent with the results depicted in Fig. 7b,c. (<50% and <60%, respectively).
Response:
We appreciate the reviewer’s careful observation. The inconsistency arose from a mislabeling in the initial version of the manuscript. The values of 93.1% and 69.2% were mistakenly cited from preliminary test data under different assay conditions. In the revised manuscript, we have corrected the text to ensure consistency with Figure 7b,c, which accurately reflect the ABTS and OH radical scavenging rates (<50% and <60%). The Results section (Lines 444) has been updated accordingly, and we sincerely apologize for this oversight.
(2) Missing of control experiments: The authors should have included a control sample (Combination of all components except GSE or with GSE replaced with a non-phenolic entity). Any literature on this?
Response:
We agree with the reviewer that such a control would strengthen the mechanistic interpretation. In our current study, we focused on comparing GSE-SeNPs with Na₂SeO₃, Vc, and GSE alone to highlight the synergistic role of GSE in reduction and stabilization. Unfortunately, we did not include a formulation in which GSE was replaced by a non-phenolic entity. However, relevant literature supports this point. For example, Salem et al. (2022) reported that orange peel waste could act as a biogenic reductant and stabilizer for SeNPs, but highlighted the critical contribution of polyphenolic constituents to nanoparticle stability and antibacterial efficacy (Salem et al., 2022). Similarly, Baluken et al. (2024) demonstrated that SeNPs synthesized with green coffee bean extracts displayed more uniform morphology and enhanced bioactivity compared to non-polyphenolic stabilizers (Baluken et al., 2024).
(3) In lines 462-463, the authors stated that: "Studies have found that SeNPs have a more significant antibacterial effect on Gram-negative bacteria compared with Gram-positive bacteria [43]." But this study showed the opposite for GSE-SeNPs. Any plausible explanation of the reversed selectivity?
Response:
We thank the reviewer for raising this important point. Indeed, many previous reports indicated stronger antibacterial activity of SeNPs against Gram-negative bacteria due to their thinner peptidoglycan layer. However, in our study, GSE-SeNPs exhibited stronger inhibition against Staphylococcus aureus (Gram-positive) than Escherichia coli (Gram-negative). Several possible explanations can account for this reversed selectivity:
Role of GSE polyphenols: GSE contains abundant proanthocyanidins and catechins with intrinsic antibacterial activity, particularly against Gram-positive bacteria by disrupting cell wall integrity. This synergistic effect may enhance the selectivity of GSE-SeNPs toward Gram-positive strains.
Surface interactions: The organic coating of GSE-SeNPs provides multiple hydroxyl and carboxyl groups that may interact more efficiently with the thick peptidoglycan layer of Gram-positive bacteria, facilitating nanoparticle adhesion and uptake.
Strain-dependent variability: Antibacterial activity of SeNPs is strongly influenced by the specific strain tested, nanoparticle size, and surface chemistry. Similar strain-dependent selectivity shifts have been reported by Galić et al.(2020), who designed SeNPs coated with polyvinylpyrrolidone (PVP-SeNP), poly-L-lysine (PLL-SeNP), and polyacrylic acid (PAA-SeNP) to obtain neutral, positively charged, and negatively charged SeNPs, respectively. All the SeNPs studied showed antibacterial activity against Gram-positive Staphylococcus aureus (24 h MBC 25-50 mg Se/L), but had no antibacterial effect against Escherichia coli and Saccharomyces cerevisiae (Galić, E. et al., 2020); In a similar study, Rangrazi et al. (2019) showed that positively charged chitosan SeNPs had good antibacterial activity against Gram-positive bacteria (Streptococcus sanguinis, Staphylococcus aureus, and Enterococcus faecalis), but had no bactericidal effect against Gram-negative bacteria (Pseudomonas aeruginosa, Salmonella typhimurium, and Escherichia coli) (Rangrazi, A. et al., 2019).
Related references:
Salem, S.; Fouda, A.; Awad, M.A.; Al-Olayan, E.M.; Allam, N.G.; Shaheen, T.I.; et al. Green Biosynthesis of Selenium Nanoparticles Using Orange Peel Waste: Characterization, Antibacterial and Antibiofilm Activities against Multidrug-Resistant Bacteria. Life 2022, 12(6), 893.
Baluken, P.; Kızıl, M.; Akdaş, S.; et al. Green Synthesis of Selenium Nanoparticles Using Green Coffee Beans: An Optimization Study. Chemistry & Biodiversity 2024, e202301250.
Galić, E.; Ilić, K.; Hartl, S.; Tetyczka, C.; Kasemets, K.; Kurvet, I.; Milić, M.; Barbir, R.; Pem, B.; Erceg, I.; et al. Impact of surface functionalization on the toxicity and antimicrobial effects of selenium nanoparticles considering different routes of entry. Food Chem. Toxicol. 2020, 144, 111621.
Rangrazi, A.; Bagheri, H.; Ghazvini, K.; Boruziniat, A.; Darroudi, M. Synthesis and antibacterial activity of colloidal selenium nanoparticles in chitosan solution: A new antibacterial agent. Mater. Res. Express 2019, 6, 1250h3.